# Fat Embolism Syndrome in Sickle Cell Disease

**DOI:** 10.3390/jcm9113601

**Published:** 2020-11-08

**Authors:** Dimitris A. Tsitsikas, Jessica Bristowe, Jibril Abukar

**Affiliations:** 1Haemoglobinopathy Service, Department of Haematology, Homerton University Hospital NHS Foundation Trust, London E9 6SR, UK; j.abukar@nhs.net; 2Research and Innovation Department, Homerton University Hospital NHS Foundation Trust, London E9 6SR, UK; jessica.bristowe@nhs.net; 3School of Pharmacy and Biomedical Sciences, University of Portsmouth, Portsmouth PO1 2UP, UK

**Keywords:** Fat Embolism Syndrome, Bone Marrow Necrosis, Sickle Cell Disease, Parvovirus B19, Exchange Transfusion, Therapeutic Plasma Exchange

## Abstract

Fat embolism syndrome is a devastating complication of sickle cell disease resulting from extensive bone marrow necrosis and associated with high mortality rates, while survivors often suffer severe neurological sequelae. Despite that, the syndrome remains under-recognised and under-diagnosed. Paradoxically, it affects exclusively patients with mild forms of sickle cell disease, predominantly HbSC and HbSβ^+^. A significant number of cases occur in the context of human parvovirus B19 infection. We provide here a brief summary of the existing literature and describe our experience treating 8 patients in our institution. One patient had HbSS, 6 HbSC and 1 HbSβ^+^. All patients developed type I respiratory failure and neurological involvement either at presentation or within the first 72 h. The most striking laboratory abnormality was a 100-fold increase of the serum ferritin from baseline. Seven patients received emergency red cell exchange and 1 simple transfusion. Two patients (25%) died, 2 patients (25%) suffered severe neurological impairment and 1 (12%) mild neurological impairment on discharge, while 3 (38%) patients made a complete recovery. With long-term follow-up, 1 patient with severe neurological impairment and one patient with mild neurological impairment made dramatic improvements, making the long-term complete recovery or near complete recovery rate 63%. Immediate red cell exchange transfusion can be lifesaving and should be instituted as soon as the syndrome is suspected. However, as the outcomes remain unsatisfactory despite the increasing use of red cell exchange, we suggest additional therapeutic measures such as therapeutic plasma exchange and pre-emptive transfusion for high risk patients.

## 1. Introduction

Fat embolism syndrome (FES) is a well described complication of long bone fractures and orthopaedic surgery characterised by the development of respiratory failure, neurological involvement, skin rashes and thrombocytopenia [1]. Non-traumatic FES is rare and is the result of extensive bone marrow necrosis (BMN) [2]. The first case of FES in sickle cell disease (SCD) was reported in 1941 [3], with several more cases reported since. 

BMN results in the release of fat globules as well as haematopoietic tissue in the venous circulation. Circulating phospholipids are metabolised to arachidonic acid under the influence of secretory phospholipase A_2_ (sPLA_2_), which in turn gives rise to a number of inflammatory cytokines. A degree of limited BMN with release of fat droplets in the circulation is probably a chronic phenomenon in SCD, as evidenced by raised sPLA_2_ levels even at steady state in SCD patients compared to normal individuals [4], while during vaso-occlusive crises (VOC) release of fat from the necrotic marrow can embolise the lungs, leading to the acute chest syndrome (ACS) [5]. When BMN is extensive, it can result in the release of large amounts of fat droplets that enter the systemic circulation giving rise to FES and the characteristic picture of multiorgan dysfunction [6]. The deleterious effects of fat embolism are the consequence of both mechanical obstruction and tissue toxicity through the formation and action of inflammatory cytokines [4,7].

## 2. The Literature

In 2014, we undertook the largest systematic review of all cases of FES in SCD reported in the literature [8], with an update of that review published in 2019 [9]. Collectively, we identified 87 cases. Thirteen (15%) patients had homozygous disease (HbSS), while 43 (50%) had HbSC and 16 (18%) HbSβ^+^ (Figure 1). As the reported distribution of SCD per genotype in patients of predominantly African origin in the Americas and the UK is ~75% for HbSS, 18–24% for HbSC and <5% for HbSβ^+^ with an anticipated genotype-related severity SS > SC/Sβ^+^ [10,11], the observed reversal of proportionate representation of FES cases Sβ^+^ > SC > SS indicated that milder forms of SCD carried unexpectedly the highest risk for developing this syndrome. This was further supported by the fact that the vast majority of cases, including the HbSS patients, had previously a mild clinical course of their disease with more than one third of cases having previously undiagnosed SCD.

Even though some patients presented with evidence of end-organ failure, sometimes in extremes, the most typical presentation was that of a seemingly uncomplicated VOC of unusual severity followed by the rapid development of respiratory failure, neurological impairment, fever, thrombocytopenia and evidence of other organ dysfunction.

Evidence of recent infection with human parvovirus B19 (HPV B19) was documented in 15 out of the 67 (22%) of cases reported after 1986. This is very likely to be an underestimate as only 10 (15%) had a documented negative result with the remainder not making any mention of HPV B19 testing. From 3 cases published in 2019, 1 tested positive, 1 tested negative (but positive for HSV 1) and 1 was not tested at all while in a case series published in 2020, 4 of 7 patients (57%) tested positive for HPV B19 [12,13,14,15]. 

Overall mortality in cases published until 2013 was 66% and reduced to 33% in cases from 2014–2018. Mortality was greatly affected by the use of transfusion and was 23%, 59% and 92% for patients receiving red cell exchange (RCE), simple transfusion (ST) and no transfusion respectively. Severe neurological impairment (SNI) was seen in 12 (14%) patients surviving FES; it should be noted that all 12 cases were published after 2012. In fact, we have previously observed that the reduction in mortality from 66% to 33% in cases published after 2014 was accompanied by a dramatic increase in cases of SNI from 5% before 2014 to 33%. Complete recovery (CR) was seen in only 21%, and this figure was the same before and after 2014. Patients with HbSS, despite representing only 15% of all cases of FES, accounted for 28% of patients making a complete recovery, indicating that such patients not only have a lower risk of developing FES, but may also suffer less as a result of it.

## 3. Our Experience

To date, we have treated 8 adult patients with SCD and FES in our institution presenting from 2012 to 2019 and we have previously published some data on 3 of them [16]. Their characteristics were identical to those observed in the cases we found reported in the literature: 1 patient had HbSS, 6 HbSC and 1 HbSβ^+^. Up to that point, all patients had a very mild course of their illness, with very infrequent or no hospital presentations for management of VOC or any other complication. The only evidence of end-organ damage was the presence of avascular necrosis (AVN) for 2 patients (Table 1).

### 3.1. Clinical Features

All patients presented with a painful VOC of unusual—for them—severity. Two patients had type I respiratory failure at presentation while 1 had *E. Coli* sepsis. Subsequently, all patients developed severe type I respiratory failure and varying degrees of neurological or other systems involvement. The evolution of the clinical picture of the patients involved is summarised in Table 2.

### 3.2. Laboratory Investigations

A detailed description of various haematological and biochemical parameters and their evolution during the course of the presentation can be found in Table 3. The single marker that was universally markedly deranged at presentation with a 100-fold increase of its mean value from baseline was the serum ferritin. Lactate dehydrogenase (LDH) was markedly elevated at presentation for 7 of the 8 patients. Subsequently, very high LDH levels were observed for all patients. Even though all but one patients had normal alkaline phosphatase (ALP) levels at presentation, ALP was significantly increased in all 8 patients during the course of their presentation. Three patients had ALP isoenzyme analysis and that showed marked predominance of the bone isoenzyme. Two patients were thrombocytopenic at presentation but eventually all patients dropped their platelet (PLT) count from a mean 246 × 10^9^/L to 110 × 10^9^/L.

### 3.3. Imaging

Plain chest radiographs were performed before the development of respiratory failure for 3 patients and were unremarkable. Those performed (or repeated) when patients developed type I respiratory failure showed diffuse bilateral infiltrates for 6 while the other 2 plain x-rays were unremarkable. Bedside echocardiography performed for those 2 patients showed markedly elevated pulmonary artery pressures. Five patients had brain imaging by computed tomography (CT) and magnetic resonance imaging (MRI); even though CT was unremarkable for all, diffusion-weighted imaging by MRI revealed widespread punctate microhaemorrhages in 4 of the 5, giving the characteristic “star-field” pattern.

### 3.4. Other Investigations

The peripheral blood smear of all 8 patients showed a leucoerythroblastic picture with a high number (26–42%) of circulating nucleated red blood cells (nRBC). However, this picture was not evident for all patients from presentation and serial blood films were required to capture the evolution. Of the 7 patients who had a bone marrow biopsy (including one autopsy), only 3 had clear evidence of BMN. It should be noted that the other 4 had a typical clinical picture, including 2 patients with characteristic brain changes by MRI. Two patients (25%) had evidence of recent HPV B19 infection, 4 were negative and 2 not tested.

### 3.5. Management and Outcomes

All patients were managed in the intensive care unit (ICU), 6 requiring intubation and mechanical ventilation at some point and one requiring haemofiltration. Seven received emergency RCE (2 manual, 5 automated), while the eighth had two units of red cells but an erroneous diagnosis of pulmonary oedema delayed RCE by 24 h after developing respiratory failure, by which time the condition was terminal and the patient succumbed before RCE was commenced. Two patients died (25%), 2 suffered SNI (25%), 1 mild neurological impairment (12%) and 3 patients (38%) made a complete recovery with no further sequelae 3, 5 and 7 years from presentation respectively. One of the 2 patients with severe neurological impairment after prolonged rehabilitation improved remarkably and apart from some mild memory problems has returned to normal 5 years after the original event. Equally, the patient with mild neurological impairment (left arm hemiparesis) also recovered fully approximately 2 years from presentation. Taking that under consideration, 5 of the 8 patients (63%) achieved CR or near-CR eventually.

### 3.6. Reflection

Even in an institution where we have a special interest in FES with regular presentation of cases and teaching, diagnosis of one case was delayed by 24 h, while two cases were not tested for HPV B19. Even severe neurological sequelae may improve dramatically over time and access to intensive rehabilitation is of paramount importance.

## 4. Diagnosis and Management

The only features universally present at onset are the characteristic patient profile of a previous mild course of SCD (most often non-homozygous) and a markedly elevated serum ferritin. All patients develop type I respiratory failure, but this often does not manifest until hours or even days after presentation. The same applies to involvement of the nervous or other systems and the development of thrombocytopenia and other haematological and biochemical abnormalities. Bone marrow biopsy can lend support to the diagnosis but cannot exclude it, as in our experience typical cases of FES did not show evidence of extensive BMN on biopsy; this may reflect heterogeneous marrow involvement or, more likely, dependence on the timing of sampling due to rapid recovery of the bone marrow after the acute event.

There are established diagnostic criteria for traumatic FES [17,18,19]. However, we agree with other investigators [20] that these do not apply to FES in the context of SCD, as other common complications of the disease such as stroke or sepsis can have a similar picture. Development of specific diagnostic criteria for FES in SCD would require prospective validation, which is unlikely to be feasible given the paucity of cases. Moreover, survival of patients depends on immediate institution of treatment as soon as FES is suspected and several clinical or laboratory diagnostic features may not be present early, when a clinical decision is required. FES should be seriously considered in any patient fitting the characteristic genotypic/phenotypic profile presenting with evidence of end-organ dysfunction and/or markedly elevated inflammatory markers and/or thrombocytopenia.

For documentation and reporting purposes, more “descriptive” definitions as previously published [8] may be more appropriate: FES was defined as multi- or single-organ histologically proven involvement by fat and/or necrotic marrow emboli or development of acute respiratory distress and neurological manifestations or multiorgan failure with evidence of BMN (pathological proof or laboratory evidence). BMN was defined as histologically proven (autopsy or biopsy) extensive BMN, radiological evidence, or a relevant clinico-pathological picture; that is bone pain and development of anaemia and thrombocytopenia with a leucoerythroblastic peripheral smear with high numbers of circulating nRBCs [21].

### 4.1. Red Cell Exchange Transfusion 

The evidence from the literature and our own experience clearly indicate that RCE can be lifesaving and should be instituted as soon as FES is suspected. Our practice is to transfuse to Hb S or Hb S&C of less than 10–20%. Whenever possible, automated RCE should be employed as it is much more effective in rapidly reducing the sickle level. Investigations that may lend further support to the diagnosis should certainly be undertaken but under no circumstances delay transfusion.

### 4.2. Therapeutic Plasma Exchange 

Despite the unquestionable benefit from urgent RCE, the outcomes remain poor, with a high mortality rate and a large number of surviving patients left with significant neurological impairment. We have previously advocated the use of therapeutic plasma exchange (TPE) following adequate RCE [9]. The rationale is that TPE can potentially remove fat droplets and harmful circulating cytokines produced from phospholipids released from the necrotic bone marrow. TPE has been previously used successfully to clear circulating cytokines in cases of macrophage activation syndrome or “cytokine storm” syndrome complicating rheumatological conditions, as well as in the context of acute liver failure for the same purpose [22]. There are several case reports and case series on the safe use of TPE in patients with SCD both in the context of multiorgan failure [23], as well as in the context of microangiopathic haemolytic anaemia (MAHA)/thrombotic thrombocytopenic purpura (TTP), with excellent clinical outcomes [24,25,26,27,28]. Interestingly, all cases reported as TTP also had severe respiratory failure, which is not a feature of TTP, all cases received red cell transfusion before TPE, and ADAMTS 13 levels were not tested at all or tested after treatment and found to be normal. The risk of misdiagnosis of FES as TTP has previously been reported [29,30,31], raising the possibility that these cases not only demonstrate that TPE is safe in SCD but also that many of these cases reported as TTP and treated successfully with RCE followed by TPE may have indeed been cases of FES. TPE has also been used successfully after RCE for management of ACS [32].

We have so far used TPE in two patients with FES. One patient (HbSS) made a complete recovery while the other (HbSβ^+^) died on day 3 of TPE. There were no adverse events that we could directly attribute to the use of TPE. We have also used TPE as monotherapy in a patient with severe acute sickle cell intrahepatic cholestasis that could not receive red cell transfusion due to a previous life-threatening transfusion reaction; TPE led to rapid and complete recovery (in press). A group at the University of Alabama at Birmingham successfully treated a patient originally referred with a diagnosis of TTP which was revised to FES [9], while another group at UCLA also treated a patient with HPV B19 positive FES using RCE followed by 5 cycles of TPE, leading to complete neurological recovery (personal communication). Our current policy is to perform 5 cycles of TPE, replacing 1 plasma volume/procedure using solvent detergent processed plasma (Octaplas) as replacement fluid, monitoring clinical response as well as the levels of ferritin, LDH and platelets. Depending on those, further TPE procedures may be considered. As the value of TPE is not yet tested adequately, patients (or relatives) are counselled regarding the rationale of using it while pointing out the lack of significant evidence on its benefit before consenting.

### 4.3. Pre-Emptive Red Cell Exchange Transfusion

Given the rapid evolution of the syndrome often leading to death or severe neurological damage, in our institution we also consider pre-emptive (before any evidence of organ failure) RCE for high risk patients. The advantage of such an approach is that it may halt the process before leading to extensive BMN and FES. Ironically, this is also its main disadvantage as it will be impossible to tell in retrospect whether at risk patients did not develop FES as a result of the intervention. In any case, pre-emptive RCE requires careful counselling and explaining the rationale but also the risks of transfusion in the absence of clear evidence of clinical benefit before patients consent. Our diagnostic and management schematic algorithm can be seen in Appendix A.

In our institution, we are developing a technique for detection of a range of inflammatory cytokines from exhaled breath condensate (EBC) which could provide a quick non-invasive bedside test that may help identify at presentation patients at high risk of developing ACS or FES and may benefit from early intervention.

### 4.4. Long Term Management

For patients achieving CR, we are currently not recommending any further intervention. For patients with neurological or other sequelae, we are recommending chronic RCE or, alternatively, treatment with hydroxycarbamide. As mentioned earlier, for such patients we have seen neurological recovery over time, so participation in specialist rehabilitation programs is essential.

## 5. The Elusive Pathogenesis

The pathophysiological process causing extensive bone marrow necrosis resulting in systemic fat embolism in patients with sickle cell disease is not known. The paradox of this most severe complication affecting exclusively patients with mild forms of the disease raises suspicion that the pathogenesis may be linked precisely to their unique biological profile. Unlike “typical” homozygous patients, these patients are expected to retain some splenic function, the ability to build up immune responses may suggest an immune mediated mechanism. This is further supported by the observed association with HPV B19 infection. HPV B19 is known to cause immune mediated phenomena in the general population from very common and mild such as erythema infectiosum and arthralgias to uncommon but well described severe, even life-threatening immune mediated complications [33,34,35,36,37,38,39]. In SCD patients, HPV B19 infection typically induces transient red cell aplasia (TRCA) through direct cellular toxicity to erythroid progenitors. Other complications such as acute glomerulonephritis, acute splenic sequestration and cerebrovascular events have also been described and are also the result of direct cellular toxicity or the consequence of severe anaemia in the context of TRCA [40,41,42,43,44]. Interestingly, immune mediated complications such as those observed in the general population have not been described in SCD [45,46]. Conversely, there are no reported cases of HPV B19-induced bone marrow necrosis in the general population. The population of SCD patients we see affected by FES may share characteristics with the two groups above, and be prone to immune mediated complications (like the general population) due to retention of some splenic function but also prone to bone marrow necrosis due to abnormal bone marrow stroma and microvasculature [47] (like typical homozygous SCD patients), (Figure 2). Even though we clearly see an association between FES and HPV B19 infection, a plethora of other infectious or non-infectious immunogenic stimuli may have the same effect. FES in SCD has been reported in the context of HSV-1 infection [13] and after different drug administration [48,49,50,51]. Additionally, a retrospective analysis of the outcomes of dengue fever in 70 children with SCD has shown that patients with HbSC and a mild SCD phenotype were worse affected than homozygous patients, with a higher incidence of multiorgan failure and higher mortality; this phenomenon was hypothesised to be the result of immune mediated endothelial dysfunction [52]. Finally, the impact of COVID-19, a virus shown to have the potential of inducing brisk immune responses with devastating effects [53], on patients with SCD appears to be significantly more severe for patients with mild forms of the condition. Data from 83 patients in France [54], 178 patients in the US [55] and 195 patients in the UK [56] consistently showed that patients with mild manifestations of SCD are worse affected, with a higher rate of intensive care admissions and higher case fatality for patients with HbSC/Sβ^+^ compared with homozygous patients.

Putting this or other theories to the test can be a fascinating field of research as not only will it lead to better understanding and improved management of the syndrome, but it also has the potential of shedding light on other aspects of SCD biology. Patients with the mildest forms of SCD being the ones exclusively affected by its most catastrophic complication presents us with an unexpected paradox. Studying such “outliers” is what often reveals new mechanisms of disease.

## 6. Conclusions

The management of fat embolism syndrome in sickle cell disease is a race against time. The profile of patients developing this complication and the clinical presentation is characteristic and almost identical amongst affected individuals, aiding early recognition and timely institution of therapeutic measures as long as clinicians consider the syndrome. We feel there is a pressing need for a multicentric exploration of new therapeutic approaches such as TPE or pre-emptive RCE as the outcomes remain poor despite increasing use of red cell exchange transfusion.

## Figures and Tables

**Figure 1 jcm-09-03601-f001:**
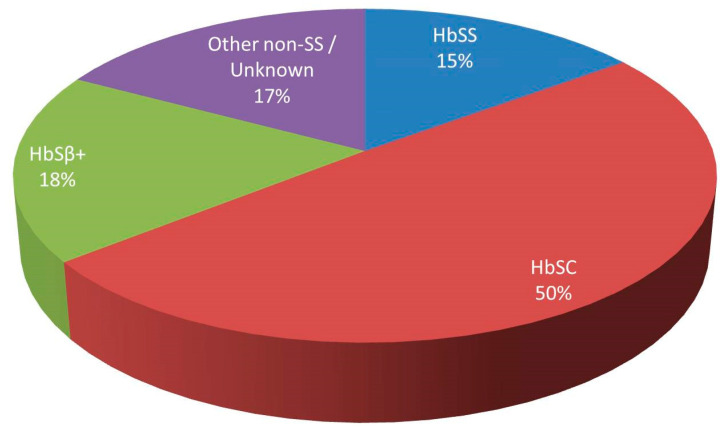
Distribution of fat embolism syndrome cases by sickle cell genotype.

**Figure 2 jcm-09-03601-f002:**
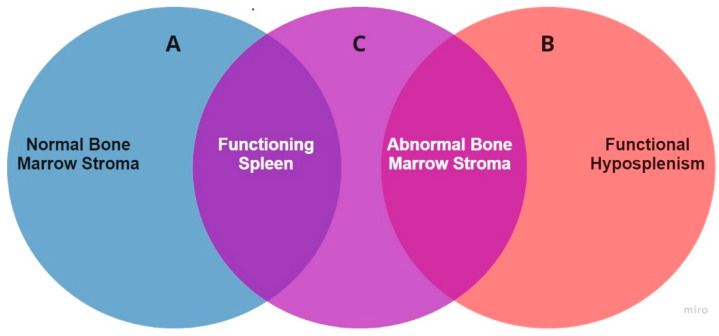
Overlap of biological characteristics between the general population (**A**), “typical” homozygous sickle cell patients (**B**) and patients with sickle cell disease at risk of developing fat embolism syndrome (**C**).

**Table 1 jcm-09-03601-t001:** Patient characteristics, management and outcomes.

Pt	GT	BMN D	FES D	HPV B19	PD	Mx	OC On DC/On FU
1	SC	CL	CL/MRI	Y	MD/AVN	RCE	SNI/nCR
2	SS	BMBx	CL	NT	MD/AVN	RCE/TPE	CR
3	SC	CL	CL	Y	MD	RCE	CR
4	SC	CL	CL	NT	MD	RCE	CR
5	SC	BMBx	CL/MRI	N	MD	RCE	MNI/CR
6	SC	A	A	N	MD	ST	D
7	SC	CL	CL/MRI	N	MD	RCE	SNI/SNI
8	Sβ^+^	BMBx	CL	N	MD	RCE/TPE	D

GT = genotype, BMN D= bone marrow necrosis diagnosis, FES D = fat embolism syndrome diagnosis, PD = previous disease, MD = mild disease, Mx = management, OC = outcome, DC = discharge, FU = followup, CL = clinical, A = autopsy, BMBx = bone marrow biopsy, Y = yes, N = No, NT = Not tested, RCE = red cell exchange transfusion, TPE = therapeutic plasma exchange, ST = Simple transfusion, D = death, SNI = severe neurological impairment, MNI = mild neurological impairment, CR = complete recovery, nCR = near complete recovery.

**Table 2 jcm-09-03601-t002:** Clinical features at onset and evolution (hours from presentation) during admission.

Pt	Respiratory Failure (Hours)	Fever (Hours)	Neurology (Hours)	>50% PLT (Hours)	>50% Cr (Hours)	Liver imp. (Hours)	Other (Hours)	LOS (Days)
1	25	25	Coma 32	37	No	No	No	407 (334)
2	48	0	Altered level of consciousness 32	31	47	Severe hyperbilirubinaemia 84	*E. Coli* sepsis 0	65
3	23	No	Altered level of consciousness 23	0	0	Severe transaminitis 23	Skin lesions 48	34
4	0	0	3rd branch V nerve sensory loss 0	24	30	Mild transaminitis 0	No	14
5	0	12	Left arm Hemiparesis 28	0	28	Mild transaminitis 20	Hypertension 0	126
6	22	22	Altered level of consciousness 22	27	34	Severe transaminitis 22	No	2 RIP
7	60	56	Coma 60	No	60	No	No	211 (160)
8	71	No	Altered level of consciousness 71	71	71	Severe transaminitis 71	No	7 RIP

Clinical features and time from presentation (Hours): > 50% PLT: Reduction by 50% or more of platelets from baseline; >50% Cr: Increase by 50% or more of creatinine from baseline: Figures in brackets indicate days in rehabilitation after the acute presentation.

**Table 3 jcm-09-03601-t003:** Haematological and biochemical parameters at onset and evolution (hours from presentation) during admission.

	Baseline Mean (Range)	Presentation Mean (Range)	Max/Nadir Value Mean (Range)	Max/Nadir Hours Mean (Range)
**Ferritin mcg/L**	109 (10–214)	10,119 (2274–40,000)	18,172 (2274–50,014)	42 (0–76)
**LDH U/L**	298 (126–630)	2167 (678–5287)	3516 (1105–6852)	15 (0–37)
**CRP mg/L**	<5	52 (<5–235)	317 (196–493)	102 (38–198)
**Haemoglobin g/L**	113 (75–130)	108 (69–136)	86 (45–116)	31 (7–75)
**Reticulocytes × 10** ^**9**^ **/L**	121 (75–165)	127 (78–196)	43 (12–81)	105 (7–267)
**PLT × 10** ^**9**^ **/L**	246 (150–330)	236 (85–338)	110 (31–231)	46 (0–75)
**Creatinine umol/L**	70 (48–90)	82 (74–146)	284 (82–936)	84 (37–140)
**ALP U/L**	77 (48–100)	138 (43–410)	475 (283–774)	115 (16–246)
**ALT U/L**	21 (11–35)	88 (18–309)	2475 (18–11,262)	131 (0–397)
**Bilirubin umol/L**	23 (8–65)	40 (10–119)	156 (18–766)	66 (0–153)

Haematological/biochemical parameters and progression from presentation (Hours): The nadir platelet count is considered the lowest value before exchange transfusion as automated exchange transfusion can cause thrombocytopenia.

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
