# Peer review of "Fat Embolism Syndrome in Sickle Cell Disease"

_jcm, 2020, doi:10.3390/jcm9113601_

Round 1
Reviewer 1 Report
I read with interest and anticipation this report considering that I also consider FES in sickle cell disease a very important topic that needs to be better known and recognized.
I have the following suggestions:
- Summarize the data from the literature in a table. It will make it much easier to compare the findings between the various periods of time (until 2013 versus 2014-2018, for example).
- Table 2 is truncated in the PDF. I am not sure what is missing from the right side. Also, please clarify how the "hours" are counted - I assume from the presentation to the hospital, right?
- Page 10, line 193 - I would replace "sickle level" with "hemoglobin S or SC". SIckle level is mentioned again at the end of line 194 and it sounds too coloquial.
- Having treated at least 20 patients with this syndrome, I completely disagree with the sentence that starts in line 196 (... invariably needs to be repeated to maintain...). Considering that red cells have such a long circulating life, I have never had to repeat the exchange transfusion. Simple transfusion may be necessary, but it would be mainly to correct symptomatic anemia. Please delete this sentence to avoid sending the wrong message.
- When referring to therapeutic plasma exchange, please use the ASFA abbreviation TPE, instead of TPEX.
- Please delete "cycles of TPEX" with "TPE procedures". A cycle is a group of TPEs.
- Page 13, line 257 - the sentence that starts with "Specifically, as unlike ...", is confusing. Please rewrite it.
- Page 10, line 261: the word "manifestations" should be added after "mild".
- Several times, the authors use red cell exchange transfusion - transfusion is not appropriate. Red cell exchange is enough to refer to the procedure.
- Finally, please state that TPE for FES has not been tested enough - thus, it cannot be recommended at this time.
Author Response
Thank you for reviewing our manuscript and your valuable comments. We found them very useful and constructive.
Please see below our response to your comments:
ï‚· Summarize the data from the literature in a table. It will make it much easier to compare the findings between the various periods of time (until 2013 versus 2014-2018, for example).
DAT: We have avoided doing this in the first place as precisely such a table exists in our 2019 publication which is cited in the text. Even though some duplication of the previously published work is unavoidable, we tried to avoid it as much as possible while pointing the readers to the original papers. Having said that, if the editors consider it necessary we are happy to include the table from our 2019 paper in the British Journal of Haematology.
ï‚· Table 2 is truncated in the PDF. I am not sure what is missing from the right side. Also, please clarify how the "hours" are counted - I assume from the presentation to the hospital, right?
DAT: I am afraid this is the result of the tables being on a Word document using a “landscape” layout while the main document is “portrait”. The Journal may be able to assist you by making the Word document available to you.
"hours" does indeed mean from presentation and this is now made clear in the Legends.
ï‚· Page 10, line 193 - I would replace "sickle level" with "hemoglobin S or SC". SIckle level is mentioned again at the end of line 194 and it sounds too coloquial.
DAT: This has now been addressed
ï‚· Having treated at least 20 patients with this syndrome, I completely disagree with the sentence that starts in line 196 (... invariably needs to be repeated to maintain...). Considering that red cells have such a long circulating life, I have never had to repeat the exchange transfusion. Simple transfusion may be necessary, but it would be mainly to correct symptomatic anemia. Please delete this sentence to avoid sending the wrong message.
DAT: This has now been addressed
ï‚· When referring to therapeutic plasma exchange, please use the ASFA abbreviation TPE, instead of TPEX.
DAT: This has now been addressed
ï‚· Please delete "cycles of TPEX" with "TPE procedures". A cycle is a group of TPEs.
DAT: This has now been addressed
ï‚· Page 13, line 257 - the sentence that starts with "Specifically, as unlike ...", is confusing. Please rewrite it.
DAT: This has now been addressed
ï‚· Page 10, line 261: the word "manifestations" should be added after "mild".
DAT: This has now been addressed
ï‚· Several times, the authors use red cell exchange transfusion - transfusion is not appropriate. Red cell exchange is enough to refer to the procedure.
DAT: This has now been addressed
ï‚· Finally, please state that TPE for FES has not been tested enough - thus, it cannot be recommended at this time.
DAT: We have now added that our policy is to counsel patients regarding the rationale of using TPE while pointing out the lack of significant evidence on its benefit and at the same time at our Conclusion we add a statement regarding the need for multicentric exploration of novel approaches such as TPE.
Kind regards
Dr Dimitris A. Tsitsikas

Reviewer 2 Report
Bone marrow necrosis and fat embolism syndrome are rare complications occuring in patients with sickle cell disease and are different from those which happen in traumatic conditions
This work is very important for physicians who manage these patients because of the very low incidence and high level of fatal issue of these complications. Reliable diagnosis and efficient management are difficult.
Thus, the authors'aim, for several years, is to gather retrospectively updated and reliable information on clinical and biological presentation of these complications and their management, from their own experience as well as from others. These results because of the scarcity of these complications are difficult to achieve by prospective approach.
Indeed, they propose an algorithm for early recognition and management of fat embolism syndrome in adult patients with sickle cell disease.
They show that Patients profile has to be taken into consideration. It seems that they occured mainly in patients with history of previous mild sickle cell disease.
Viral infection or other situations (pregnancy, steroids,..) as for our experience with dengue fever, seem to be significant triggers. Multicentric study is required to identify the pathophysiology of this phenomenon
Early treatment with Red cell exchange associated with the low risk therapeutic plasma exchange are proposed by the authors. these treatment has to be tested in a multicentric procedure to test their efficiency in significant reduction of fatal issue
Others authors described few cases
Tsitsikas et al. work significantly improves knowledge on diagnosis of bone marrow necrosis and fat embolism syndrome in sickle cell disease. The authors proposed a therapeutic approach which has to be tested with much more patients by a multicentric approach.
Others studies are also required to validate the algorithm (multicentric studies)
Author Response
Thank you for reviewing our manuscript and your valuable comments. We found them very useful and constructive.
Kind regards
Dr Dimitris A. Tsitsikas
Reviewer 3 Report
The paper from Tsitsikas et al describes the case series of 8 SCD patients experiencing fat embolism syndrome. The paper is well written and the topic is of interest. Relevant physical examination, laboratory testing and interventions are described in the text and summarized in 2 tables (not completely readable due to a wrong layout), both clear and comprehensive with the help of the legend.
The time frame of the study is not completely clear to me.
Please provide a brief sentence stating that the patients gave their informed consent for publication.
Have any of the 8 cases been described from the authors previously? Such as in the case series published in the Clin Case Rep. 201 7 Jan; 5(1): 39–43?
Author Response
Thank you for reviewing our manuscript and your valuable comments. We found them very useful and constructive.
Please see below our response to your comments:
The paper from Tsitsikas et al describes the case series of 8 SCD patients experiencing fat embolism syndrome. The paper is well written and the topic is of interest. Relevant physical examination, laboratory testing and interventions are described in the text and summarized in 2 tables (not completely readable due to a wrong layout), both clear and comprehensive with the help of the legend.
DAT: I am afraid this is the result of the tables being on a Word document using a “landscape” layout while the main document is “portrait”. The Journal may be able to assist you by making the Word document available to you
The time frame of the study is not completely clear to me.
DAT: We have now added the time span these patients were treated.
Please provide a brief sentence stating that the patients gave their informed consent for publication.
DAT: Even though we have consent from all 6 living patients we felt it was inappropriate and insensitive to approach their relatives of the two deceased patients 2 and 4 years after the tragic event when there are no patient identifying data. This was approved and endorsed in writing by our institutional R&I department.
Have any of the 8 cases been described from the authors previously? Such as in the case series published in the Clin Case Rep. 201 7 Jan; 5(1): 39–43?
DAT: Indeed. This is now mentioned and referenced in the manuscript.
Kind regards
Dr Dimitris A. Tsitsikas